# An Internationally Derived Process of Healthcare Professionals’ Proactive Deprescribing Steps and Constituent Activities

**DOI:** 10.3390/pharmacy12050138

**Published:** 2024-09-09

**Authors:** Sion Scott, Natalie Buac, Debi Bhattacharya

**Affiliations:** 1School of Healthcare, University of Leicester, Leicester LE1 7RH, UK; s.scott@leicester.ac.uk; 2School of Pharmacy, University of East Anglia, Norwich NR4 7TJ, UK; n.buac@uea.ac.uk

**Keywords:** polypharmacy, shared decision making, behaviour change, intervention, deprescription

## Abstract

Proactive deprescribing is the process of tapering or stopping a medicine before harm occurs. This study aimed to specify and validate, with an international sample of healthcare professionals, a proactive deprescribing process of steps and constituent activities. We developed a proactive deprescribing process framework of steps which we populated with literature-derived activities required to be undertaken by healthcare professionals. We distributed a survey to healthcare professionals internationally, requesting for each activity the frequency of its occurrence in practice and whether it was important. Extended response questions investigated barriers and enablers to deprescribing. The 263 survey respondents were from 25 countries. A proactive deprescribing process was developed comprising four steps: (1) identify a patient for potential stop of a medicine, (2) evaluate a patient for potential stop of a medicine, (3) stop a medicine(s), and (4) monitor after a medicine has been stopped, and 17 activities. All activities were considered important by ≥70% of respondents. Nine activities required healthcare professionals to undertake in direct partnership with the patient and/or caregiver, of which seven were only sometimes undertaken. Deprescribing interventions should include a focus on addressing the barriers and enablers of healthcare professionals undertaking the activities that require direct partnership with the patient and/or caregiver.

## 1. Introduction

Deprescribing is the process of tapering or stopping medicines to minimise unnecessary polypharmacy and improve medicine-related outcomes [1]. Deprescribing may be undertaken reactively in response to harm caused by a medicine or proactively in order to prevent a medicine causing harm in the future [2]. Reactive deprescribing requires relatively simple decision making and is widely accepted as being routine practice [2,3,4]. Proactive deprescribing is a component of Good Prescribing Practice [5], which requires prescribers to regularly evaluate, in partnership with the patient, the ongoing suitability of the medicines that they prescribe. Routine proactive deprescribing aims to prevent harm and avoids the need to reactively deprescribe medicines.

Trials of proactive deprescribing have demonstrated that it is safe and leads to positive outcomes including reductions in polypharmacy, falls, and admissions to hospital [6,7,8]. Despite the recognised benefits and development of numerous interventions to increase healthcare professionals’ proactive deprescribing behaviour [9], it is not routine practice [2,10,11]. Whilst approximately 50% of older adults admitted to hospital are prescribed at least one medicine that should be stopped, an evaluation of 2309 older adults in the United Kingdom hospital setting found that less than 1% had a medicine proactively deprescribed [2].

There is a substantial body of research describing healthcare professionals’ barriers and enablers (determinants) to proactive deprescribing. A systematic review reported that knowledge gaps, insufficient time, and perceived resistance from patients are barriers [3]. Proactive deprescribing is a complex process and, for it to be undertaken safely, healthcare professionals must undertake a series of activities. Some activities may already be routinely undertaken, for example, taking a medication history on admission to hospital, whilst others may be more challenging and require interventions to support healthcare professionals in undertaking them [12].

Avoiding redundancy when developing interventions is essential in order to minimise complexity and focus resources efficiently [13]. There is a need to identify and define the activities that healthcare professionals find most challenging when undertaking proactive deprescribing. This permits the diagnosis of the barriers and enablers of these challenging activities and subsequently the development of interventions that address them.

The full breadth of activities that healthcare professionals are required to undertake for proactive deprescribing are yet to be established. Numerous deprescribing processes which provide overarching steps have been proposed; however, there is substantial heterogeneity in content, and none describes the constituent activities that healthcare professionals should undertake [14]. Processes describe one or a combination of the following steps [14]: (1) take medication history, (2) identify inappropriate medicines, (3) evaluate the suitability of deprescribing, (4) plan and deprescribe, and (5) monitor. Whilst these overarching steps provide a high-level indication of what should be achieved during the proactive deprescribing process, there remains a need to specify the activities within these steps that healthcare professionals should undertake. Examples include specifying activities that require the healthcare professional to work in partnership with the patient and/or informal caregiver to undertake deprescribing that is borne out of shared decision-making [15]. Once the full breadth of activities have been specified, there is a need to establish which are already routinely undertaken and which require targeting by interventions.

This study aimed to specify and validate, with an international sample of healthcare professionals, a proactive deprescribing process of steps and constituent activities. We also aimed to estimate the extent to which the sample of healthcare professionals perceived that activities are and are not usual practice within their peer group and, for those that are not usual practice, to identify the determinants that require addressing by deprescribing interventions.

## 2. Materials and Methods

### 2.1. Developing a Draft Proactive Deprescribing Process of Steps and Activities

We worked with our stakeholder group (*n* = 8) of UK-based patients and caregivers with lived experience of deprescribing and doctors, nurses and pharmacists who were independent of the research team and whose clinical role included deprescribing.

We developed a proactive deprescribing process framework comprising the following five overarching steps derived from 10 articles [16,17,18,19,20,21,22,23,24,25] identified from an existing literature review of deprescribing processes by Reeve et al. [14]: (1) take medication history (2) identify inappropriate medicines (3) evaluate the suitability of deprescribing (4) plan and deprescribe, and (5) monitor. We then populated the framework with the activities required to be undertaken by healthcare professionals in order to fulfil each step. With guidance from the stakeholder group, we also indicated which activities require healthcare professionals to undertake in direct partnership with the patient or caregiver.

Two researchers (DB and PW) independently reviewed the 10 articles [16,17,18,19,20,21,22,23,24,25] from which the five overarching steps were derived and extracted descriptions of the deprescribing activities required to be undertaken by healthcare professionals. We mapped activity descriptions to the relevant deprescribing step based on their relatedness. Any similar or overlapping activities were combined. We compared mapping between researchers and harmonised terminology between activities, e.g., care recipient, patient and older adult were harmonised to patient. We refined the mapping and terminology by seeking feedback from the stakeholder group. This resulted in a draft process of proactive deprescribing steps and constituent activities.

### 2.2. Validation of the Proactive Deprescribing Process of Steps and Activities

We validated the draft process using an online survey distributed to an international audience of healthcare professionals across several health systems.

#### 2.2.1. Survey Development

We produced the online survey incorporating items inviting respondents to indicate, on a 5-point Likert scale from ‘never’ to ‘always’, how often they thought each deprescribing activity from the draft process was currently undertaken. Items were phrased to ask respondents to respond based on what they thought their peer group did, rather than their own behaviour, to reduce the chance of social desirability bias [26]. For each activity, we included a second item to capture the extent to which respondents felt that an activity was sufficiently important to be part of the deprescribing process. The response options were on a 3-point Likert scale of ‘not important’, ‘unsure’, and ‘important’. We included an item asking respondents to indicate whether they used any tools or guidelines to support them in identifying inappropriate medicines. Finally, we included two global items: the first inviting respondents to indicate how often they thought their peers successfully involved patients/caregivers in the process of deprescribing and the second inviting respondents to indicate how successful they themselves were in deprescribing on a scale of 1 (never successful) to 10 (always successful). This was followed by two extended response items inviting healthcare professionals to report any barriers and enablers to them undertaking the deprescribing activities.

We piloted and refined the survey with the stakeholder group in order to establish face validity.

#### 2.2.2. Survey Distribution

Ethical approval was secured from the University of East Anglia Faculty of Medicine and Health Sciences Research Ethics Committee (Reference: 2020/21-020) to undertake an international cross-sectional survey of healthcare professionals whose role includes deprescribing. An invitation to complete the survey was distributed internally via relevant organisations and networks including our social media, professional bodies, and networks representing healthcare professionals. Members of these organisations were eligible if they were a doctor, nurse, pharmacist, or other healthcare professional whose role involved prescribing. The survey was hosted on Microsoft^®^ Forms (Appendix A) from November 2020 to April 2021. It comprised an embedded information leaflet and electronic confirmation of consent. Non-identifiable demographic information to enable us to characterise the respondent population was captured.

### 2.3. Sample Size Justification

This study did not aim to detect any differences between respondent groups; thus, no formal sample size calculation was performed. We used the convention of setting minimum and maximum target sample sizes for survey responses of 100 and 1000 responses, respectively [27]. Based on 50% of participants selecting the midpoint value (i.e., 5) for the global deprescribing success item, 100 responses provided a 95% confidence interval (CI) of ± 9.8%, thus providing acceptable precision [27,28].

### 2.4. Analysis

We calculated descriptive statistics using IBM SPSS version 27 to characterise the respondent population and report survey responses. For each deprescribing activity, we calculated the percentage of respondents indicating that it was important. We considered an activity as important if ≥70% of respondents rated it as important [29]. We calculated the median and interquartile (IQ) frequency with which each deprescribing activity was reported to be currently undertaken and for the global item regarding successfully involving patients/caregivers in the process of deprescribing. For respondents’ personal perceived level of deprescribing success on the 10-point scale, we report the mean and 95% CI.

We performed a content analysis on the extended response data in Microsoft^®^ Excel, coding for barriers to and enablers of deprescribing [30]. The barriers and enablers were mapped to the relevant domain of the Theoretical Domains Framework (TDF) by a research pharmacist (NB); mapping was checked by researchers experienced in using the TDF (SS and DB) [4].

## 3. Results

### 3.1. Developing a Draft Process of Proactive Deprescribing Steps and Activities

We extracted 17 activities from the 10 articles [16,17,18,19,20,21,22,23,24,25], 9 of which were deemed by our stakeholder group to require direct partnership with the patient/caregiver. Appendix B provides the data extracted from the 10 articles and the activities these underpinned. Based on feedback from our stakeholder group, we refined the deprescribing process framework by combining Step 1 (take medication history) and Step 2 (identify inappropriate medicines) into identifying a patient for potential stop of a medicine. This was in recognition of the stakeholders’ view that activities within these steps are often undertaken simultaneously in practice. They also recommended minor rephrasing of the other three steps.

Table 1 provides the four overarching deprescribing steps, the 17 constituent activities, and the articles from which the activities were derived. Appendix B provides the data underpinning the activities that we extracted from each article. Three articles contributed five activities [17,24,25], one article contributed four [20], four articles contributed two [16,18,22,23], and two articles contributed a single activity [19,21].

### 3.2. Validation of the Proactive Deprescribing Process of Steps and Activities

Of the 263 survey respondents, 110 (41.8%) were doctors, 85 (32.3%) were pharmacists, 44 (16.7%) were nurses, and 24 (9.1%) were other healthcare professionals. Most of the doctors (87 (79.0%)) were hospital specialists, and the remainder were family physicians. Europe accounted for 186 (69.9%) respondents spanning 10 countries, 26 (9.9%) respondents were from two countries in North America, five (2.0%) were from four countries in South America, six (2.4%) were from six countries in Asia, four (1.5%) were from Egypt, eight (2.4%) were from Australia and one was from New Zealand.

All 17 deprescribing activities met the threshold of ≥70% of respondents considering them important. Figure 1 provides the reported frequency of deprescribing activities being undertaken within the respondents’ peer group. The only activity with a median score of 5, indicating that it was perceived to always happen within the peer group of respondents, was ‘asking the patient/relative to disclose all medications they are taking’. Seven of the nine activities to be undertaken in direct partnership with the patient/caregiver were only ‘sometimes’ undertaken. The activity perceived to be undertaken with the lowest frequency with a lower quartile of respondents indicating that it occurs rarely was ‘asking the patient/relative about their thoughts and experiences of taking their medication’.

In step 2 of evaluating a patient for the potential to stop a medicine, 160 (60.8%) respondents cited a tool or resource to help evaluate the likelihood of harm or benefit from a medicine.

The median (IQ) for the global item inviting respondents to indicate how often they thought their peers successfully involved patients/caregivers in the process of deprescribing was 3 (3, 4), indicating that they perceived the majority of their peers to only sometimes achieve this outcome.

The mean (95% CI) of 7.13 (±0.19) on a scale of 1 to 10 for the perceived level of deprescribing success indicated that respondents perceived themselves to be successful in deprescribing in over two-thirds of situations where they felt that a medicine was inappropriate and should be stopped.

The coded barriers to and enablers of deprescribing obtained from the extended response data were mapped to 13 of the 14 TDF domains, with only ‘Optimism’ being absent. Two enablers and four barriers were consistently reported across the data and are summarised in Figure 2. ‘Social influence’ and ‘Environmental context and resources’ appear both as barriers and enablers that require addressing. Conversely, ‘Knowledge’ and ‘Skills’ do not require addressing with interventions because respondents indicated that they already have the knowledge to identify any medicines that should be deprescribed and to safely deprescribe them. They also have the skills to navigate deprescribing consultations with patients to achieve a shared decision.

Respondents recognised that at the point of initiation, some medicines are presented to patients as ‘lifelong’. This practice contributed to the ‘Social influence’ barrier of a perception that patients were likely to be reluctant to have a medicine deprescribed that had been presented as lifelong when first prescribed.

“*Patients may have been started and told they need to be on [the medicine] for life … They may also worry that their condition will deteriorate, and this is more of a concern than potential side effects that may or may not occur.”*Pharmacist

‘Social influence’ as a barrier also arose from peers with conflicting goals and priorities associated with prescribing for a given patient, particularly when this was reinforced by therapeutic area prescribing guidelines.

“*Most challenging are colleagues of other specialties who in some instances are reluctant to stop some medications that are potentially harmful*.” Doctor

‘Social influence’ from patients as an enabler was supported by establishing a good rapport through effective communication of deprescribing opportunities and involvement in decision making. Respondents described how the patient then became a partner, rather than someone who needed convincing that deprescribing was necessary. Similarly, adopting a collaborative approach to working with peers who may have conflicting goals was a proposed solution.

“*Having a multidisciplinary team approach as well as patient engagement. Good communication between primary and secondary care is also important*.” Pharmacist

The barrier related to ‘Environmental context and resources’ recognised that deprescribing is a resource-intensive process, with respondents reflecting on the struggle to find sufficient time to undertake all 17 activities. Enablers proposed were dedicated time to deprescribe and strategies that make the process more efficient, for example, using the Screening Tool of Older Person’s Prescriptions (STOPP) criteria [31] to make the step of ‘Evaluating a patient for potential stop of a medicine’ more efficient.

“*I need sufficient time, resources and ability to stop medicines successfully on a daily basis*.”Pharmacist

## 4. Discussion

The 17 activities that make up the four steps provide a specification validated by an international sample of healthcare professionals for the process of proactive deprescribing. Just over half of the activities were perceived to be regularly undertaken by respondents. Activities requiring direct partnership with the patient or caregiver represented all of those that were only sometimes undertaken. Proactive deprescribing interventions are required to facilitate increased delivery of these activities [15]. Interventions should comprise components that address the four barriers and two enablers to facilitate healthcare professionals to undertake these patient-facing activities. Interventions should also motivate healthcare professionals to reflect on the extent to which they themselves successfully involve patients and/or caregivers in the relevant activities.

The doctor, pharmacist, and nurse study participants represented the mainstay of healthcare professional groups involved in deprescribing [3] and a range of healthcare systems spanning 25 countries. Their validation that all 17 activities are important provides content validity. Contrary to previous deprescribing processes [14], activities associated with both undertaking a medication history and evaluating whether a medicine should be stopped are combined into one step (Step 1). This was in recognition of stakeholders’ view that activities within these steps are often undertaken simultaneously in practice. This step is phrased ‘Identifying a patient for potential stop of a medicine’, reflecting the intention that the process guides service design and intervention development.

Every step of the proactive deprescribing process comprises one or more activities requiring direct partnership with the patient and/or caregiver. This aligns with the principles of shared decision making, which define success as incorporating ‘what matters most’ to the patient in all stages of healthcare processes [15]. It is therefore of concern that respondents thought that their peers only sometimes successfully involved patients/caregivers in the deprescribing process. In contrast, respondents felt that in the majority of cases, they had successfully deprescribed a medicine that they perceived inappropriate. Targeting interventions to support healthcare professionals to involve the patient and/or caregivers across the four deprescribing steps has the potential to increase appropriate proactive deprescribing and enhance the patient and caregiver experience.

Several barriers to engaging with patients and caregivers provide explanations for the vast majority of relevant activities only sometimes being undertaken in practice. Perceived patient reluctance to have a medicine deprescribed was prominent. A substantial body of literature using the revised Patient Attitudes Towards Deprescribing (rPATD) questionnaire consistently reports that >90% of patients and caregivers are hypothetically willing to have a medicine deprescribed [32]. This suggests that the perceived patient reluctance expressed by respondents may be a misconception. Conversely, several large deprescribing trials have reported that the majority of patients, in fact, decline proactive deprescribing when it is proposed to them [33,34], aligning with our respondents’ perceptions. This disconnect between patients indicating a high hypothetical willingness to have a medicine deprescribed in response to the rPATD and subsequently declining when it is proposed in practice may be explained by the patient-oriented activities in the proactive deprescribing process developed in this study not being undertaken routinely, leading to an unsatisfactory patient experience and a subsequent unwillingness to have a medicine deprescribed.

The perception that patients are resistant to deprescribing stems from healthcare professionals failing to prime them for the possibility of deprescribing [3]. For medicines prescribed by a specialist in particular, there is an ‘unspoken rule’ not to intervene with their prescribing, leading to a heightened perception of patient resistance [3]. More generally, perceived resistance was explained by patients being told that medicines are ‘for life’ when they are first prescribed. Even prescribers recognising this as problematic practice went on to self-report that they themselves sometimes do not convey that deprescribing is a future possibility at the point of initiating a medicine. The rationale for this practice was not explored in this study, but it may be explained by the emphasis placed on influential treatment guidelines on medicines being for the ‘long-term’ [35], a recognised strategy to encourage medication adherence [36].

Lack of time is a well-recognised barrier to proactive deprescribing [3], and this study identified that the activities that succumb are those requiring engagement with patients. These activities are the most time-consuming and least flexible in terms of when they can be undertaken. Consultations are time-limited, particularly in primary care, and both patients and healthcare professionals report that they prioritise non-deprescribing activities such as history taking, diagnosis, and prescribing [3,37]. Whilst other activities can be incorporated around healthcare professionals’ competing priorities, these activities can only be undertaken when the patient or a caregiver is available. Dedicated proactive deprescribing clinics are a potential solution [38]; an Australian pilot study demonstrated that a clinic for older people frequently admitted to hospital was feasible and acceptable [39]. A hospital stay or care home setting may present opportunities to undertake these activities given that patients and/or caregivers are more available than in other settings [7,37]. However, these alternative settings may present other challenges such as patients being acutely unwell or lacking the capacity to engage.

Consistent with the existing literature [3], this sample of healthcare professionals perceived themselves as having the required knowledge and skills to proactively deprescribe. Despite this, two-thirds reported making use of deprescribing tools, which are interventions widely used to address knowledge and skills gaps [40]. This disconnect between respondents reporting adequate knowledge and also using tools which provide knowledge [41] may suggest that healthcare professionals are using these tools for reasons other than knowledge. They may be using these tools as a prompt and cue to deprescribe [40] or a heuristic to facilitate efficient deprescribing decisions [40,42]. The incorporation of prompts and cues into deprescribing interventions has been extensively studied. The SENATOR randomised controlled trial demonstrated that prompts and cues incorporated into electronic prescribing systems did not increase deprescribing behaviour [11,43]. Healthcare professional participants reported that prompts and cues were burdensome, disruptive, and prone to alert fatigue [43]. There is significant evidence beyond the field of deprescribing asserting that tools serve as effective aids to heuristics [44,45]. There is a need for strategies to ensure that deprescribing tools are more likely to serve as heuristics to facilitate ‘Memory attention and decision making’, which may be achieved by co-designing their implementation with the target audience [46,47,48].

After a medicine has been deprescribed, respondents reported that their peer group ‘often’ monitors patients for adverse withdrawal effects. Adherence to monitoring of clinical outcomes such as blood pressure or hospitalisation is unsurprising given that these are often requirements laid out in guidelines [35] and therefore subject to organisational auditing and benchmarking [49]. However, monitoring patients’ quality of life being only sometimes undertaken suggests that healthcare professionals are not considering patient-centred outcome measures. This may be because monitoring of these outcomes is not routine and requires a paradigm shift towards a greater focus on quality of life and patient goals. Another explanation may be the dominance of ‘reactive’ deprescribing behaviour in current practice [2], given that clinical outcomes such as resolution of a gastric bleed upon deprescribing anticoagulants are intuitive to monitor under these circumstances, whilst improving patient-centred outcomes is not the primary aim of reactive deprescribing [50].

The international reach and representations from the primary healthcare professional groups involved in deprescribing are a strength of this study which affords confidence in the validity of the resulting deprescribing process and its constituent steps and activities. Given that recruitment was facilitated by professional networks and social media, it was impossible to establish the response rate for this study. However, it is likely that the 263 survey respondents represent a small proportion of the population who were invited to participate and thus introduces the risk of self-selection bias [51], particularly in favour of healthcare professionals who were already confident in deprescribing. However, recognition that all but one of the patient-orientated deprescribing activities were sub-optimally delivered in practice affords some confidence that the findings provide an approximate reflection of real practice. Moreover, the UK-based stakeholder group evaluated and confirmed face validity in the UK context, which likely extends to other similar English-speaking countries. However, the extent to which the survey had face validity in other contexts, especially those in which English is not the main language or whose health systems are substantially different, is unclear and is therefore a potential limitation. Healthcare professionals have affirmed that all 17 proposed deprescribing activities are essential and that, whilst some of these are already routinely undertaken in practice, activities requiring engagement with patients are sub-optimally delivered. To develop efficient interventions, developers should focus on including components targeting the eight activities requiring healthcare professionals to engage with patients. The four barriers and two enablers have been mapped to the TDF, which, through its linkage to relevant behaviour change techniques, provides a range of theory-informed potential intervention components to support healthcare professionals in undertaking the activities requiring engagement with patients. The resultant intervention may lead to a substantial change in the patient’s experience with deprescribing. Whilst increased engagement with patients in deprescribing may be welcomed by patients [37], trials testing interventions should ensure that they capture the patient experience of deprescribing to inform refinement and evaluation.

## Figures and Tables

**Figure 1 pharmacy-12-00138-f001:**
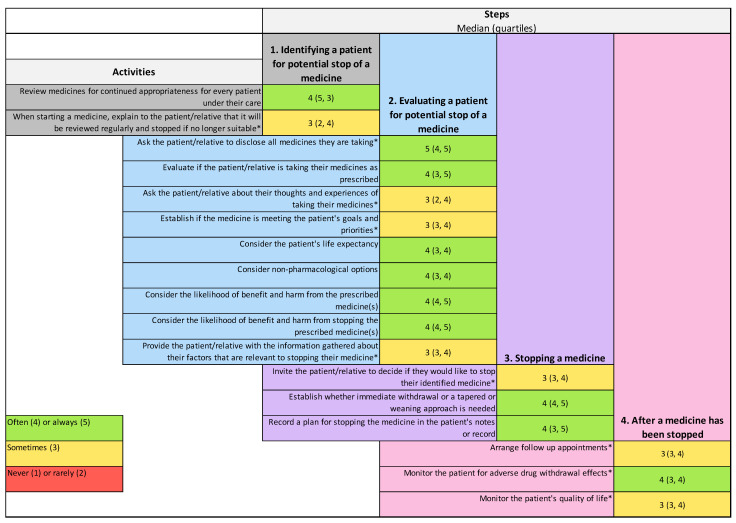
Reported frequency of deprescribing activities within respondents’ peer group. Frequencies are reported as medians (quartiles): 1 = never happens, 2 = rarely happens, 3 = sometimes happens, 4 = often happens, and 5 = always happens. * Activity requires healthcare professionals to undertake it in direct partnership with the patient or caregiver.

**Figure 2 pharmacy-12-00138-f002:**
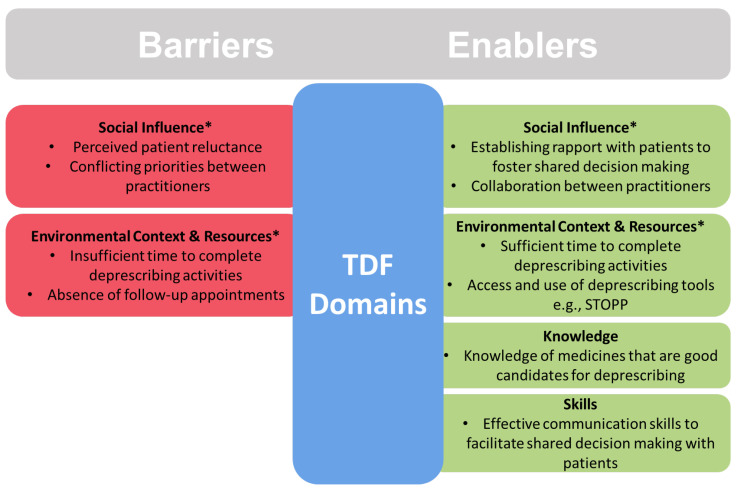
Key barriers and enablers to deprescribing within their Theoretical Domains Framework (TDF) domains. * Requires addressing with interventions.

**Table 1 pharmacy-12-00138-t001:** Draft process of proactive deprescribing steps and activities.

Proactive Deprescribing Step	Activities
Step 1. Identify a patient for potential stop of a medicine	Review medicines for continued appropriateness for every patient under their care [17,19,22,25] When starting a medication, explain to the patient/relative that it will be reviewed regularly and stopped if no longer suitable * [21,23]
Step 2. Evaluate a patient for potential stop of a medicine	Ask the patient/relative to disclose all medications they are taking * [16,20,24,25] Evaluate if the patient/relative is taking their medication as prescribed [25] Ask the patient/relative about their thoughts and experiences of taking their medication * [25] Establish if the medication is meeting the patient’s goals and priorities * [20,24] Consider the patient’s life expectancy [20,24] Consider alternative non-pharmacological options [16] Consider the likelihood of benefit and harm from continuing to prescribe the medicine(s) [17,18,24] Consider the likelihood of benefit and harm from stopping the medicine(s) [17] Provide the patient/informal caregiver with the information gathered about their factors that are relevant to stopping their medicine * [25]
Step 3. Stop medicine(s)	Invite the patient/informal caregiver to decide if they would like to stop their identified medicine * [22] Establish whether immediate withdrawal or a tapered or weaning approach is needed [23] Recording a plan for stopping the medicine in the patient’s notes or record [17,18]
Step 4. After a medicine has been stopped	Arrange follow-up appointments * [20] Monitor the patient for adverse drug withdrawal effects * [17] Monitor the patient’s quality of life * [24]

* Activity must be undertaken by healthcare professionals in direct partnership with the patient/caregiver.

## Data Availability

The raw data supporting the conclusions of this article will be made available by the authors upon request.

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
