# Peer review of "An Internationally Derived Process of Healthcare Professionals’ Proactive Deprescribing Steps and Constituent Activities"

_pharmacy, 2024, doi:10.3390/pharmacy12050138_

Round 1

Reviewer 1 Report

Comments and Suggestions for Authors

Please see Word file

Author Response

Reviewer 1

Comment 1

The abstract refers to describing as stopping (1/10) but the introduction definition states

tapering or stopping (1/28). Should tapering be added to abstract?

Response

We have added tapering to the abstract.

Comment 2

The authors (1/22) state ‘should focus on’. Would ‘should include a focus on’ be more

aligned with study, considering the sample and method of recruitment to the study?

Response

We have amended the text as recommended in the abstract.

Comment 3

There are a number of questions where participants are asked about ‘usual practice

within your peer group’ (Appendix A); it would be helpful if the wording of the second

aim (2/74) would more explicitly align with the wording within questions.

Response

Thank you for the suggestion to refine the aim; we have amended as recommended to:

We also aimed to estimate the extent to which the sample of healthcare professionals perceived that activities are and are not usual practice within their peer group, and for those that are not usual practice, to identify the determinants that require addressing by deprescribing interventions.

Comment 4:

UK-based patients, carers and professionals (n=8) formed the stakeholder group

establishing lived-experiences which formed part of the development of the framework

in addition to review of articles (2/79). This formed the basis of establishing face-

validity for the pilot (the same 8 as I read the paper). These three points should be

acknowledged in the discussion as limitations.

Response

Thank you for this suggestions, we have added the following text to the discussion:

Moreover, the UK-based stakeholder group evaluated and conformed face validity of the survey in the UK context, which likely extends to other similar English-speaking countries. However, the extent to which the survey had face validity in other contexts, especially those for which English is not the main language or whose health systems are substantially different, is unclear and is therefore a potential limitation.

Comment 5

I notice the name of the ethics committee was redacted but the names and affiliations

of the author were clear on page 1 (2/123)

Response

We have added the redacted information:

Ethical approval was secured from the University of East Anglia Faculty of Medicine and Health Sciences Research Ethics Committee (Reference: 2020/21-020) to undertake an international cross-sectional survey of healthcare professionals whose role includes deprescribing.

Comment 6

Should the words read ‘1 (never successful)’ ? (3/116) rather than ‘never unsuccessful’/

Response

Thank you for spotting this error; we have corrected.

Comment 7

What do the colours signify in Appendix B? I footnote would be helpful (4/159)

Response

Apologies for the confusion regarding the colours; they were intended to distinguish between the four steps however when uploading it seems some of the colours were replaced with the Excel table blue. We have corrected this and hopefully this is now much clearer.

Comment 8

Should these three lines be deleted? (8/261-263)

Response

We have deleted these.

Comment 9

Authors used ‘they’ in parts where the respondents themselves rather than the peer

group. It may be clearer to readers to refer to ‘they themselves’ when referring to their

own practice in the manuscript. For example (9/290), I think I recall there being another

/other instance/s.

Response

Thank you; we agree this requires clarifying and have amended various points in the manuscript with ‘they themselves’ to distinguish between when we are referring to a perception of respondents’ peer groups’ practice versus their own practice.

Comment 10

It would be helpful for the authors to indicate what findings ‘suggest’ use of tools for

other purposes. Was it the discussions of stakeholder group? (9/338)

Response

We have clarified the foundation of this discussion point, which is us making a potential link between respondents saying they have sufficient knowledge ‘already’ to deprescribe, yet they use tools which provide knowledge on how to deprescribe in their practice. We have softened the language as this is a hypothesis rather than a definitive finding, as follows:

Consistent with the existing literature[3], this sample of healthcare professionals perceived themselves to have the required knowledge and skills to proactively deprescribe. Despite this, two-thirds reported making use of deprescribing tools, which are interven-tions widely used to address knowledge and skills gaps[40]. This disconnect between re-spondents reporting adequate knowledge and also using tools which provide knowledge[41] may suggest that healthcare professionals are using these tools for reasons other than knowledge. They may be using these tools as  a prompt and cue to depre-scribe[40] or a heuristic to facilitate efficient deprescribing decisions[40,42].

Comment 11

I note the comments in the final paragraph (10/382-386). However, when one considers

the number of respondents, and they were self-selecting, and the method of

recruitment. Can the authors justify the following words in their conclusion? I suggest

alternate wording below

‘will’ (382) to ‘may’

‘likely to be’ (384) to ‘may be’

Response

Thank you; we have actioned.

Comment 12

Appendix B (p29) please see comment relating to (4/159, above).

Response

We have amended – please see response to comment 7.

Comment13

Figures

Would the authors consider reviewing the font sizes and also contrast of fonts with

background colours to make the contrast more accessible for readers?

Response

Thank you; we have adapted the figure accordingly.

Reviewer 2 Report

Comments and Suggestions for Authors

Report on “An internationally-derived process of healthcare professionals’ proactive deprescribing steps and constituent activities”

Manuscript Number: pharmacy-3150238

pharmacy

Summary

The article evaluates a proactive deprescribing process of steps and constituent activities by surveying an international sample of healthcare professionals.

Assessment

The aim of this article is to fill an important research gap, namely the specification of concrete activities within the deprescribing process in order to develop interventions. While there is consensus on the general proactive deprescribing process, there is still a need to delineate the specific activities that healthcare professionals should undertake within each step. Further, barriers and enablers were identified. Therefore, the findings of this study enrich the empirical evidence and might guideline the development for specific interventions.

The data set used provides a valid approach to address the research question. The presentation of the results is clear and easy for the reader to follow.

In what follows, I give a couple of suggestions that may improve the paper.

Results

·         Please report the response rate.

·         Did you check for differences between professions in the frequencies of specific de-prescribing steps (Figure 1)? These results may provide interesting insights for the development of interventions for specific professions.

·         Similarly, are there differences across the considered countries?

Author Response

Comment 1

Please report the response rate.

Response

The survey was distributed openly via networks and social media and therefore it is not possible for us to report the response rate and we cannot ascertain how many people the invitation to complete the survey reached. We recognise that this is a limitation and have included the following text in the discussion:

Given that recruitment was facilitated by professional networks and social media, it is impossible to establish the response rate for this study. However, it is likely that the 263 survey respondents represents a small proportion of the population who were invited to participate and thus introduces the risk of self-selection bias[51], particularly in favour of healthcare professionals who are already confident deprescribes. However, recognition that all but one patient orientated deprescribing activity are sub optimally delivered in practice affords some confidence that the findings provide an approximate reflection of real practice.

Comment 2

Did you check for differences between professions in the frequencies of specific de-prescribing steps (Figure 1)? These results may provide interesting insights for the development of interventions for specific professions.

Similarly, are there differences across the considered countries?

Response

We did not check for these differences as the aim of the study was not to seek to explore differences between the various healthcare professional groups whose role includes deprescribing. We agree however that this would be interesting for a future study.